# The Role of Resorcinolic Lipids of Caryopsis Surface in the Process of Cereal Infection by *Rhizoctonia solani* and *Fusarium culmorum*

Elżbieta G. Magnucka *, Małgorzata P. Oksińska and Stanisław J. Pietr

Laboratory of Biogeochemistry and Environmental Microbiology, Department of Plant Protection,
Wrocław University of Environmental and Life Sciences, 50-375 Wrocław, Poland;
malgorzata.oksinska@upwr.edu.pl (M.P.O.); stanislaw.pietr@upwr.edu.pl (S.J.P.)
* Correspondence: elzbieta.magnucka@upwr.edu.pl

**Abstract:** Cereal caryopses are rich in 5-*n*-alk(en)ylresorcinols, antimicrobial compounds. In this paper, the correlation between the presence of resorcinolic lipids on the surface of cereal grains and the susceptibility of their seedlings to infection by *Rhizoctonia solani* or *Fusarium culmorum* was evaluated. The declines in length of both the roots and coleoptiles were observed in barley seedlings of Scarlett and Rabel cultivars grown from the wax-impoverished seeds infected with F92 and F93 strains of *Rhizoctonia solani*, respectively. Regarding wheat, *R. solani* F93 significantly reduced only the coleoptile growth. Resorcinolic lipids, being the mixture of homologues with C17–C25 carbon chains, were the only compounds washed off wheat caryopses by chloroform. Moreover, the better anti-*Rhizoctonia solani* F93 activity of 5-*n*-alk(en)ylresorcinols of wheat grains than that of rye caryopsis lipids was proven by the poisoned medium technique. Two saturated homologues (C21:0 and C23:0) were the most effective inhibitors of the mycelial growth of this fungus. Thus, the susceptibilities of barley and wheat seedlings to some fungal pathogens have been found to be related to the content and composition of 5-*n*-alk(en)ylresorcinols in the waxy layer of cereal grains, confirming the protective role of these compounds, during the early stages of cereal development.

**Keywords:** 5-*n*-alk(en)ylresorcinols; caryopsis; cereal surface; *Rhizoctonia solani*



## 1. Introduction

The seedling stage is considered to be one of the phases most sensitive to external stimuli in the plant life cycle. Those seedlings are susceptible in the soil to physical injuries, and thus to microbial infections, especially to fungal decay [1]. As with all plants, cereals produce protective antimicrobial secondary metabolites, either as part of their normal program of growth and development (phytoanticipins) or in response to a pathogen attack or stress (phytoalexins). Phytoanticipins, so-called preformed antimicrobial compounds, are biosynthesized constitutively in healthy plants. These substances form a chemical barrier against attack by various phytopathogenic organisms [2].

Resorcinolic lipids and their derivatives are found in cereal plants in both germinated caryopses and their seedlings [3–5]. Sun et al. [6], for example, showed the presence of alkylresorcinol synthase in all above-ground organs of rye. However, mature, ungerminated grains are the richest source of these secondary metabolites [3,4]. Interestingly, 5-*n*-alk(en)ylresorcinols are located only in the outer layers of caryopses (Figure 1). They are found in both the outer and intermediate parts of the caryopsis, including the hyaline layer, testa, inner pericarp, and the cuticle of the pericarp [7–9].

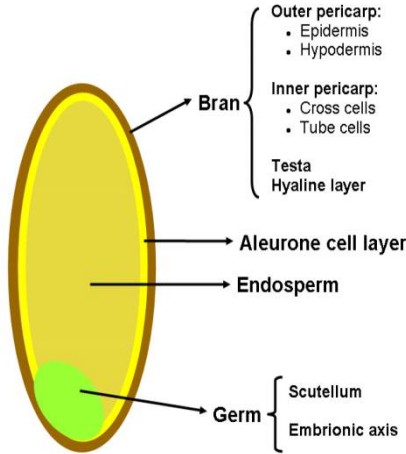

**Figure 1.** Caryopsis parts.

In general, the amount of resorcinol lipids in cereal grains is higher in rye (*Secale cereale* L.), lower in wheat (*Triticum aestivum* L.) and triticale (*Triticosecale* Wittm. Ex A. Camus), and lowest in other cereals such as barley (*Hordeum distichon* L.) and oat *Avena sativa* L.) [10–14]. A large amount of these compounds has also recently been identified in other cereals such as einkorn (*Triticum monococcum* L. subsp. *monococcum),* emmer (*Triticum turgidum* L. subsp. *dicoccum* (Schrank ex Schubl.) Thell.), spelt (*Triticum aestivum* L. subsp. *spelta* (L.) Thell.), and tritordeum (×*Tritordeum martini* A. Pujadas) [8].

Regarding their chemical nature, resorcinolic lipids, also called 5-*n*-alk(en)ylresorcinols (ARs), are long-chain, odd-numbered homologues of orcinol (1,3-dihydroxy-5-methylbenzene) that originate from the polyketide metabolic pathway [15]. These isoprenoid phenolic lipids in cereals are present as mixtures of saturated and unsaturated homologues with 13–29 carbons in the chain, including the oxidized and hydroxylated forms [10–16]. However, both the content and homologue composition of this class of phenolic lipids in cereals can vary widely between variety and species due to environmental, agricultural, and genetic factors [8,17–20]. Under in vitro conditions, the effects of various external factors, especially light, temperature, and pesticides, on the biosynthesis of these compounds have been described [21–23].

Resorcinolic lipids are potent antimicrobials and therefore they are often called natural biofungicides [24]. Hitherto, their antifungal activity, mainly against ascomycetous pathogens of the genera *Aspergillus*, *Penicillium*, and *Fusarium,* has been demonstrated [24–27].

Among several microbes affecting the development of cereal seedlings, infections caused by *Fusarium culmorum* (W.G. Smith) Sacc. and *Rhizoctonia solani* Kuhn are some of the most deleterious. *Rhizoctonia solani* is the main agent of Rhizoctonia stunt in cereals and it can be responsible for seedling rot. In turn, *F. culmorum* is a ubiquitous soil-borne fungus able to cause foot and root rot (Fusarium crown rot) and scab (Fusarium head blight; FHB) on different small-grain cereals, in particular on wheat and barley [28,29]. Interestingly, Righetti et al. [30] reported that *Fusarium* mycotoxin content, being the sum of deoxynivalenol and its glucoside, is negatively correlated with the AR level in the caryopses of *Triticum* spp.

Therefore, for the first time, the correlation between the presence of 5-*n*-alk(en)ylresorcinols on the grain surface of cereals and the sensitivity of their seedlings to infections caused by *Fusarium culmorum* and *Rhizoctonia solani* was analyzed in this work.

## 2. Materials and Methods

### 2.1. Chemicals

Solvents and the remaining reagents of analytical grade were obtained from Avantor Performance Materials Poland S.A. (Gliwice, Poland) and Merck KGaA (Darmstadt, Germany).

### 2.2. Plant Material

Grains of winter rye, cv. Dańkowskie Złote [3], winter wheat, cv. Kobra [17], and spring barley, cv. Scarlett and Rabel [18], were examined. Fully mature caryopses of the rye cultivar were provided by the "Danko" Plant Breeding Farm (Choryń, Poland). The rest of the cereals were released from the Experimental Station in Pawłowice of Wrocław University of Environmental and Life Sciences, Poland. Complete cultivar vouchers of the cereals used are available from the Central Laboratory for Studies of Cultivable Plants—COBORU (Słupia Wielka, Poland).

### 2.3. Preparation of Caryopses for a Pot Experiment

Chloroform was used to remove the waxes from the grain surface of tested cereals. Twenty grams of grains of each examined cereal cultivar (Dańkowskie Złote, Kobra, Scarlett, and Rabel) were dipped in this solvent for 15 s. The organic extract was used for further analyses, whereas the chloroform-treated caryopses, as well as the control grains (untreated with this solvent), were surface-disinfected in 5% ($w/v$) acidified chloramine-T for 10 min followed by three washes in sterile distilled water. Thereafter, these grains were placed in Petri dishes padded with moistened sterile tissue paper and allowed to germinate in the dark at 22 °C for 24 h.

### 2.4. Pot Experiment

*Fusarium culmorum* strain F1, *Rhizoctonia solani* strains F92, and F93 were chosen for this bioassay [23,31].These pathogenic fungi were grown on autoclaved ground barley grains for 14 days at 28 °C. Then, five grams of cultures were mixed with 95 grams of autoclave-sterilized sand, for each pot. Afterward, the contents of the pots were moistened with 20 mL of sterile distilled water. Two days later, the six pre-germinated grains were transplanted into a pot containing sand artificially infested with one of the fungal strains. In the control pots, five grams of the pathogen cultures were replaced with the autoclaved biomasses of the tested pathogens. The pots were incubated for 5 days with a day/night (12/12 h) temperature of 22/18 °C in a phytotron. After this time, the length of both roots and coleoptiles of cereal seedlings was measured. The experiment was replicated thrice, and three pots were considered one repetition.

### 2.5. Isolation and Determination of Total and Surface Resorcinolic Lipid Content in Cereal Grains

Alk(en)ylresorcinols were extracted from whole intact caryopses of Dańkowskie Złote, Kobra, and Rabel cultivars. Each of the 20g samples was soaked with equal volumes of acetone for 3 × 24 h. Each acetone fraction was filtered through filter paper (CHMES, Poznan, Poland). After 72 h, all acetone filtrates were combined, and the solvent was removed by vacuum evaporation on a rotavapor at 40 °C. The oily residue was redissolved in 1 mL of chloroform, and then applied to a preparative TLC plate covered with silica gel Si60 (Merck, Darmstadt, Germany). Separation was carried out in *n*-hexane/diethyl ether/formic acid (70:30:1, *v/v*). Alk(en)ylresorcinols were re-extracted overnight with a mixture of acetone/methanol (4:1, *v/v*) from a part of the gel containing these compounds. After centrifugation (7500× *g*, 10 min), the supernatant was concentrated *in vacuo* and then redissolved in methanol. The same method was also used to isolate resorcinolic lipids from chloroform fractions obtained after washing caryopses of the chosen cultivars. The micro colorimetric method was applied to estimate the level of these compounds in grains [32]. Alk(en)ylresorcinols were quantified by measuring the absorbance at 520 nm against the reagent blank. All determinations were carried out in triplicate.

### 2.6. Antifungal Activity of Wheat Caryopsis Ars

Based on the results of the pot experiment, wheat cv. Kobra and fungus *Rhizoctonia solani* strain F93 were chosen for a biotest. Resorcinolic lipids purified from the fraction obtained by the 3-day extraction of wheat grains with acetone, according to the method described above, were used to estimate their effect on mycelial growth. For this purpose, six

various doses of ARs in acetone (75, 100, 125, 150, 175, and 200 μg per mL) were loaded on the surface of solidified potato dextrose agar (PDA). This medium consists of 20 g of potato puree powder(Solan, Głowno, Poland), 2 g of BactoCasamino Acids(Becton, Dickinson and Company, Sparks, NV, USA), and 10 g of glucose in 1 L of distilled water. A medium treated with 900 μL of the solvent was used as the control. After the evaporation of acetone (1.5 h in a laminar box), a 3-mm-diameter plug of fungus grown on PDA for 3 days was placed in the center of each plate. Then, the plates were incubated at 22 °C in the dark, and every 6 h, the colony diameters of *R. solani* were measured until the fungus covered the PDA in the control plates. The percentage inhibition of radial growth of this phytopathogen was calculated relative to the growth of the acetone-treated control. This bioassay was carried out in triplicate, each with five plates as individual repeats.

### 2.7. Estimation of Concentration and Composition of Phenolic Compounds of Wheat Caryopsis Surface

The levels of total phenolics in the fraction obtained after a short chloroform wash of Kobra wheat grains were determined by the Folin–Ciocalteu (FC) method described by Singleton et al. [33] with slight modifications. The crude extract after complete evaporation was firstly dissolved in 2 mL of methanol, and then 100 μL of it was made up to 1 mL with methanol. Afterward, 0.5 mL of the FC reagent and 2 mL of the 20% ($w/v$) solution of anhydrous sodium carbonate were added. After incubation (60 min at room temperature in the darkness), the mixture was centrifuged ($10,000\times g$, 10 min, 4 °C). The absorbance of the samples at 760 nm against the reagent blank was read. Results were expressed as micrograms of ferulic acid equivalents per gram of grains based on the ferulic acid calibration curve. The test was carried out in triplicate.

To estimate the relative pattern of AR homologues in the outer cuticle of wheat grains according to the length of their side-chain, the extraction from the reverse-phase TLC RP-18 plate was applied as it was described in detail by Magnucka et al. [4].

### 2.8. Effect of C21:0 Homologue on the Anti-Rhizoctonic Activity of Resorcinolic Lipids of Rye Caryopses

A pure fraction of 5-*n*-alk(en)ylresorcinols isolated from rye grains during the 72-h acetone extraction according to the method described in Section 2.5 was diluted to obtain a 1000 μg mL$^{-1}$ solution. Then, the synthetic C21:0 compound (80 μg mL$^{-1}$ in acetone) was added to this solution to obtain two mixtures, one containing 120 μg of rye resorcinolic lipids and 6.5 μg of C21:0 in 1 mL, and the other mixture including 200 μg of rye ARs and 6.5 μg of the standard in 1 mL. The sample containing only ARs from rye grains (200 μg mL$^{-1}$) was the reference sample. The antifungal activity test against *R. solani* F93 was performed as described in Section 2.6.

### 2.9. Comparison of the Sensitivity of Rhizoctonia solani F93 to Derivatives of 5-n-Alkylresorcinols

Each of five saturated resorcinol compounds (C15:0, C17:0, C19:0, C21:0, and C23:0) at a dose of 50 μg mL$^{-1}$ was tested against *R. solani* F93 strain. The homologue dissolved in acetone was applied to the surface of PDA. Then the pathogen disc was placed centrally on that prepared medium. The radial growth of its mycelium was compared with the respective control according to the method described in Section 2.6.

### 2.10. Statistics

All results are the means of three independent experiments. Average mean values from each study are presented with the standard deviation. Comparisons between treatments were performed by Tukey's HSD test ($p = 0.05$).

## 3. Results and Discussion

### 3.1. The Effect of Chloroform Treatment of Cereal Caryopses on the Growth of Their Seedlings in the Presence of Phytopathogenic Fungi

In this work, the role of 5-*n*-alk(en)ylresorcinols of caryopsis surfaces in the infection process of their seedlings by *Rhizoctonia solani* and *Fusarium culmorum* was evaluated.

Moreover, only a short 15-s chloroform extraction was used to remove these phenolic compounds from the grain surface to allow them to germinate and develop into seedlings.

No significant effect of chloroform on the growth of either roots or coleoptiles of 5-day-old cereal seedlings compared to untreated controls was observed. However, considerable discrepancies in the impacts of phytopathogenic fungi on seedling growth of tested cereals were observed (Tables 1 and 2).

**Table 1.** The effect of chloroform treatment of barley grains on the growth of their seedlings in the presence of *Rhizoctonia solani*.

| Cultivar | Caryopsis/Sand Treatment | | RootLength [cm] [a] | ShootHeight [cm] [a] |
|---|---|---|---|---|
| | **Chloroform** | **Pathogen** | | |
| Scarlett | − | − | 6.60 ± 0.39 a | 5.24 ± 0.94 ab |
| | + | − | 6.58 ± 0.42 a | 5.96 ± 0.77 a |
| | − | *R. solani* F92 | 3.28 ± 0.33 b | 4.78 ± 0.64 bc |
| | + | *R. solani* F92 | 2.38 ± 0.13 c | 2.86 ± 0.62 d |
| | − | *R. solani* F93 | 2.71 ± 0.49 c | 4.02 ± 1.58 c |
| | + | *R. solani* F93 | 2.44 ± 0.29 c | 4.04 ± 0.53 c |
| Rabel | − | − | 6.83 ± 0.65 a | 6.39 ± 0.80 a |
| | + | − | 7.10 ± 0.82 a | 6.77 ± 0.79 a |
| | − | *R. solani* F92 | 2.33 ± 0.58 bc | 4.01 ± 0.77 bc |
| | + | *R. solani* F92 | 2.03 ± 0.46 b | 4.13 ± 0.93 bc |
| | − | *R. solani* F93 | 3.00 ± 0.42 d | 4.31 ± 0.69 b |
| | + | *R. solani* F93 | 2.62 ± 0.48 c | 3.87 ± 0.56 c |

[a] Mean values expressing the length of an estimated part of barley seedling ± SD obtained from three separate experiments. Values followed by different letters are significantly different at *p* = 0.05 according to Tukey's test; values of both each column and cultivar were analyzed separately.

**Table 2.** The effect of chloroform treatment of rye and wheat caryopses on the development of their seedlings in the presence of fungal pathogens.

| Cereal/Cultivar | Caryopsis/Sand Treatment | | RootLength [cm] [a] | ShootHeight [cm] [a] |
|---|---|---|---|---|
| | **Chloroform** | **Pathogen** | | |
| Rye/DańkowskieZłote | − | − | 3.59 ± 1.61 a | 8.42 ± 2.47 a |
| | + | − | 2.96 ± 1.25 a | 7.47 ± 1.93 a |
| | − | *R. solani* F93 | 1.39 ± 0.23 b | 1.69 ± 0.87 b |
| | + | *R. solani* F93 | 1.39 ± 0.38 b | 1.53 ± 0.53 b |
| | − | *F. culmorum* F1 | 1.45 ± 0.22 b | 3.13 ± 1.54 b |
| | + | *F. culmorum* F1 | 1.19 ± 0.34 b | 2.59 ±1.21 b |
| Wheat/Kobra | − | − | 5.68 ± 0.74 a | 13.17 ± 3.45 a |
| | + | − | 5.95 ± 1.18 a | 13.87 ± 1.13 a |
| | − | *R. solani* F93 | 4.31 ± 0.63 b | 12.74 ± 1.88 a |
| | + | *R. solani* F93 | 4.37 ± 0.93 b | 10.68 ± 1.62 b |
| | − | *F. culmorum* F1 | 0.43 ± 0.22 c | 3.15 ± 0.61 c |
| | + | *F. culmorum* F1 | 0.47 ± 0.25 c | 3.16 ± 1.06 c |

[a] Mean values expressing the length of the estimated part of cereal seedling ± SD derived from three separate experiments. Values followed by different letters are significantly different at *p* = 0.05 according to Tukey's test; values of both each column and cultivar were analyzed separately.

In comparison to the untreated control, the coleoptile heights of barley cv. Rabelwere decreased by both strains of *R. solani* to similar extents; on average, by 34.90%. In the case of barley cv. Scarlett, only strain F93 reduced its shoot growth by 23.28%. In turn, both strains of this fungal species were potent inhibitors of the root growth of either barley cultivar. Concerning the Scarlett cultivar, a stronger 58.94% decline was observed after the F93 application, whereas the F92 strain turned out to be more deleterious for the Rabel root system (the 65.88% growth reduction).

Removal of the outermost AR layer of barley grains of these two cultivars (Scarlett and Rabel) amplified the negative impact of *R. solani* strains on their seedlings (Table 1). This

phenomenon manifested itself in significant shortening of both root and coleoptile lengths. Interestingly, each of the cultivars reacted in this manner solely after treatment with the less pathogenic strain of *R. solani*. Strain F92 more effectively inhibited the elongation of Scarlett shoots than the root system, by 40.17 and 27.44%, respectively. In the case of the other cultivar, decreases in the length of both estimated parts of barley seedlings affected by strain F93 were very similar (12.67 and 10.21%). The fact that grains of this cultivar contain a lower amount of resorcinolic lipids than grains of the Rabel cultivar was likely responsible for this result [14]. In turn, the lack of the surface 5-*n*-alk(en)ylresorcinols did not increase the inhibition of barley seedling growth when the more pathogenic strain was applied.

In summary, it is well known that *R. solani* is harmful to plant growth. Damping-off, either pre- or post-emergence, is a common disease of seeds and seedlings of many plants. Furthermore, woody species, including conifers and broad-leaved plants, can be attacked by this pathogen [28,34]. Furthermore, Rhizoctonia stunt affects mainly barley, which is much more susceptible to this disease than other cereals [28]. The genetic variations between malting (Scarlett) and fodder (Rabel) cultivars, including 5-*n*-alk(en)ylresorcinol content in their grains, and consequently, their various in-born susceptibility to fungal diseases, seem to be responsible for our results. It is worth emphasizing that no other symptoms of plant disease on barley seedlings were observed.

One of the most aggressive pathogens for cereals, especially for wheat, is *F. culmorum* [29]. This fungus can cause seedling blight, foot rot, ear blight, stalk rot, common root rot, and other diseases of monocot and dicot plants [28]. It was observable that wheat plants were markedly more susceptible to the harmful activity of the *F. culmorum* F1strain than the *R. solani*F93strain. Compared to the control samples, *F. culmorum* significantly reduced the lengths of both their root systems and coleoptiles by 92.43 and 76.08%, respectively, and *R. solani* reduced only the root growth by 24.12% (Table 2). Interestingly, the chemical removal of wheat caryopsis waxes did not markedly change the adverse effects of *F. culmorum* on whole plants, whereas, in the case of *R. solani,* the shoots of seedlings that originated from wax-impoverished caryopses of wheat were significantly shorter, by 16.17%,than coleoptiles grown from grains untreated with chloroform (Table 2). It is likely that the level of 5-*n*-alk(en)ylresorcinols in grains of this cereal, even in those with the outer cuticle of the pericarp intact, was too low to prevent *F. culmorum* infection. This explanation is also supported by the fact that the 41.5% inhibition of the growth of *F. culmorum* mycelium was achieved by the cyclohexane extract of resorcinolic lipids isolated from durum wheat cv. Tirex at the high dose of 1100 $\mu$g mL$^{-1}$ [27]. The ability of this strain (*F. culmorum* F1) to biosynthesize these compounds may also be the reason for its high resistance to their action [31].

To estimate the antimicrobial potential of resorcinolic lipids, their total pool extracted from cereal caryopses with various organic solvents is normally used [24–27]. So far, only Garcia et al. [35] showed that 5-*n*-alk(en)ylresorcinols of epicuticular waxes of barley are responsible for the in-born resistance of this cereal to pathogenic aspergilli and penicillia. Anyhow, the influence of the removal of ARs of the cuticle of the fruit wall (pericarp) of cereal caryopses on their growth in the presence of phytopathogens has not been shown yet. Magnucka et al. [23] presented that only pyrazon-driven accumulation and modification of the homologue pattern (8-fold rise in the level of short-side chain ($\leq$C17) alk(en)ylresorcinols and only a small decline in C21:0 content) in rye seedlings grown in the darkness at 30 °C for 5 days improved their resistance to the infection by *Rhizoctonia cerealis* E. P. Hoeven but not *R. solani*.

### 3.2. Resorcinolic Lipids of Chloroform Fraction

The rapid immersion of wheat grains in chloroform removed 4.59 $\mu$g g$^{-1}$ of resorcinolic lipids from their surface, which is only 1.20% of the total pool of these compounds of wheat caryopses (Table 3). Wang et al. [9] showed that ARs of the surface of wheat grains wereca. 0.8% of their average total content in whole caryopses.

**Table 3.** The ability of chloroform to remove 5-*n*-alk(en)ylresorcinols from the caryopsis surface of various cereals.

| Plant | Cultivar | Alk(en)ylresorcinol Level [mg kg$^{-1}$] [a] | |
| --- | --- | --- | --- |
| | | **Total ARs in Grains** | **ARs in Chloroform Fraction** |
| Winter rye | Dańkowskie Złote | 588.07 ± 2.10 a | 9.24 ± 0.23 a |
| Winter wheat | Kobra | 381.85 ± 3.63 b | 4.59 ± 0.02 b |
| Spring barley | Rabel | 52.82 ± 0.39 c | 3.44 ± 0.03 b |

[a] Mean values expressing the concentration of alk(en)ylresorcinols ± SD were obtained from three independent experiments. Values followed by different letters are significantly different at $p = 0.05$ according to Tukey's test; values of each column were analyzed separately.

Moreover, the content of total phenolics in this crude organic fraction, determined using the Folin–Ciocalteu reagent, turned out to be the only AR concentration. Further analysis of this chloroform fraction showed that it consisted of resorcinolic lipids, being the mixture of the homologueswith C17–C25 carbon chains (Figure 2), with a marked predominance of C21 compound (41.28%). The presence of the C15 homologue was not detected, and the level of the C17 compound was the lowest of all identified (3.76%). This finding is similar to the results reported earlier by Wang et al. [9], who did not show the presence of short-side chain (≤C17) homologues of resorcinolic lipids extracted from the caryopsis surface of wheat.

*3.3. Anti-Rhizoctonia solani Activity of Resorcinolic Lipids*

The results of the poisoned food technique using the resorcinolic lipids isolated from grains of the Kobra cultivar also proved that this group of phenolic lipids markedly limited the development of *Rhizoctonia solani* strain F93. Wheat 5-*n*-alk(en)ylresorcinols were effective against this fungus, causing 8.93, 17.96, 29.24, 36.82, 43.65, and 52.96 percent inhibition at 75, 100, 125, 150, 175, and 200 µg mL$^{-1}$, respectively (Table 4). The ID50 value of 190.34 µg mL$^{-1}$ was calculated from a concentration–response curve of this group of compounds.

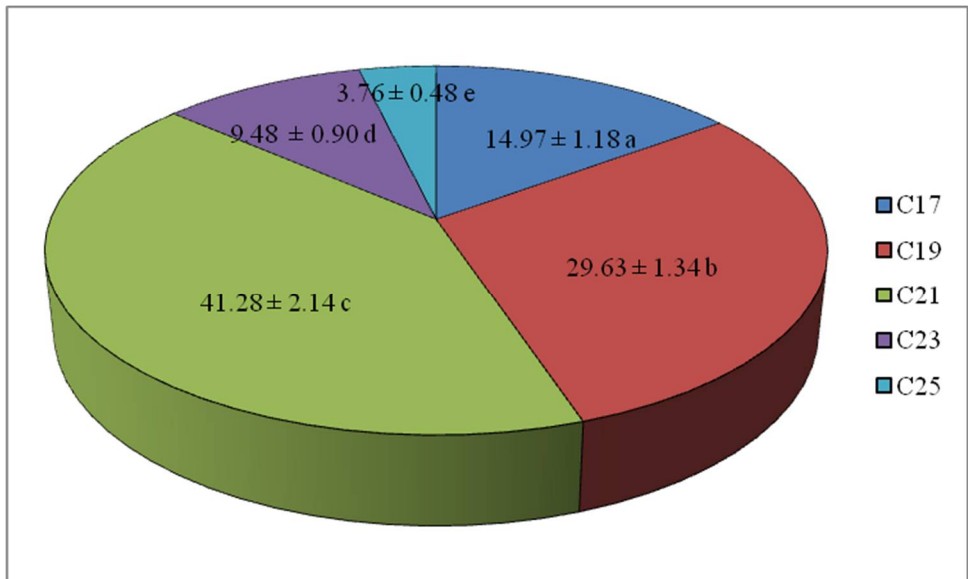

**Figure 2.** Percentage composition of 5-*n*-alk(en)yloresorcinol homologues of the wheat surface. Mean values expressing the percentage contents of alk(en)ylresorcinols ± SD were obtained from three independent experiments. Values followed by different letters are significantly different at $p = 0.05$ according to Tukey's test.

**Table 4.** The effect of wheat resorcinolic lipids on the radial growth of *Rhizoctonia solani* F93 mycelium.

| Concentrations of ARs [µg mL$^{-1}$] | Inhibition of Radial Growth [%] [a] |
|---|---|
| 75 | 8.93 ± 1.34 a |
| 100 | 17.96 ± 5.17 b |
| 125 | 29.24 ± 5.19 c |
| 150 | 36.82 ± 6.80 d |
| 175 | 43.65 ± 3.88 e |
| 200 | 52.96 ± 4.21 f |

[a] Mean values expressing the inhibition of fungal mycelium growth ± SD obtained from three analyses. Values followed by different letters are significantly different at *p* = 0.05 according to Tukey's test.

Significant growth inhibition of rye seedlings by each tested phytopathogenic fungi was observed. In comparison with control plants, the lengths of roots and shoots of rye seedlings grown in the presence of *R. solani* F93 were shortened by 61.28 and 79.93%, respectively (Table 2). Meanwhile, in the case of *F. culmorum* F1, 59.61 and 62.83% declines were accordingly noted. However, the removal of the waxy layer from its caryopses did not significantly increase the harmful effects of either of the tested fungi on the development of rye seedlings. Washing off the tiny amount of ARs (1.57%) did not change the high susceptibility of this cereal to either of the tested pathogens (Table 3). The much higher susceptibility of rye, especially to *R. solani* F93 infection, compared to other cereals was most likely related to the fact that C17:0 was the predominant homologue of its caryopses [3], whereas the C21:0 compound predominated in grains of the remaining tested cereals [12–14,17,18]. Therefore, the more efficient inhibition of *R. solani* F93 growth (approximately 2.73 times greater) by 200 µg mL$^{-1}$ of wheat grain ARs compared to the same dose of resorcinolic lipids obtained from rye grains was observed (Tables 4 and 5).However, the addition of the small amount of the C21:0 homologue (6.5 µg mL$^{-1}$) to the natural mixture of rye resorcinolic lipids almost doubled the negative effect of these phenolic compounds on the radial growth of *R. solani* F93 mycelium. Even when the level of rye resorcinolic lipids was decreased by 40%, the presence of this homologue in the mixture still significantly improved the inhibition of fungal growth by 6.37% compared to the impact of a natural extract of rye grains without this synthetic compound (Table 5).

**Table 5.** The effect of rye 5-*n*-alk(en)ylresorcinol supplementation with synthetic C21:0 homologue on the reduction of mycelial growth of *Rhizoctonia solani* F93.

| AR Content [µg mL$^{-1}$] | C21:0 Supplement [µg mL$^{-1}$] | Inhibition of Radial Growth [%] [a] |
|---|---|---|
| 200 | - | 19.41 ± 5.45 a |
| 120 | 6.5 | 25.78 ± 3.06 b |
| 200 | 6.5 | 36.97 ± 5.45 c |

[a] Mean values expressing the inhibition of fungal mycelium growth ± SD derived from three analyses. Values followed by different letters are significantly different at *p* = 0.05 according to Tukey's test.

The results described above led to an experiment in which the action of saturated homologues (C15:0–C23:0) on the growth of *R. solani* F93 mycelium was estimated. Inhibitory effects of two homologues (C21:0 and C23:0) turned out to be the most potent against this phytopathogen (Figure 3). Interestingly, the influence of these phenolic compounds, in most cases, did not differ significantly from one another and remained stable (43.05% on average) in the whole analyzed period (32 h). The negative effects of the rest of the compounds ranged from 24.15% for C15:0 to 4.72% for C17:0–C19:0.

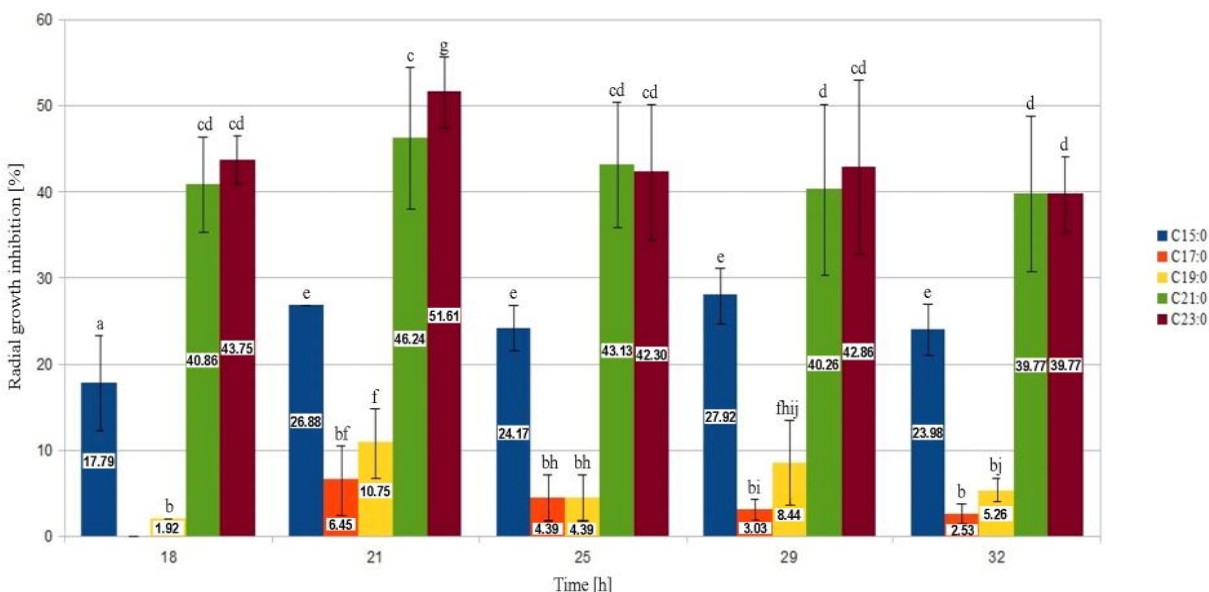

**Figure 3.** *Rhizoctonia solani* F93 growth inhibition caused by 5-*n*-alkylresorcinols (C15:0–C23:0) at 50 ppm dosage for 32 h. Means with various letters differ significantly ($p \leq 0.05$) according to Tukey's test. Bars represent $\pm$ Standard Deviation (SD).

Interestingly, the aggravation of the negative impact of the mixture of 5-*n*-alk(en)yloresorcinols, extracted from durum wheat caryopses, on *Fusarium graminearum* (Schwabe) by its enrichment with the C21:0 homologue was described previously by Ciccoritti et al. [27].However, this homologue naturally predominated in this mixture, originally representing approximately 35% of its total. In our case, this compound was added to the rye AR pool in which the natural level of this homologue was low (data not shown). Despite this fact, the increased inhibition of *R. solani* was observed. The change in the original ratio C21:0/C23:0 likely contributed. According to Ciccoritti et al. [27], the rise in this ratio increases the antifungal activity of ARs.

Thus, our results showed the essential role of the C21:0 compound, the predominant homologue in wheat caryopses, in controlling the growth of *R. solani.* Moreover, in light of these results, the lack of resistance of pyrazone-treated rye seedlings to infection by *R. solani* [23] can be explained by the decrease in the content of this homologue, which does not predominate in this cereal [3,23].

Resorcinolic lipids, as amphiphilic compounds, can incorporate into the cell membrane and increase its permeability. Gubernator et al. [36] showed that the C23:0 homologue induced higher leakage from liposomes than the C15:0 compound. In turn, Stasiuk et al. [37] reported that C21:0 was the most effective erythrocyte solubilizer of all tested saturated homologues. Thus, our findings are also in agreement with the results quoted above.

It is common knowledge that grains of barley are poor in these phenolic lipids in comparison to other cereals [14]. Our results also confirmed this observation. Rye and wheat caryopses contain over eleven- and seven-fold higher levels of 5-*n*-alk(en)ylresorcinols than barley kernels, respectively (Table 3). In addition, rapid immersion in chloroform turned out to be most effective in the disposal of 5-*n*-alk(en)ylresorcinol for barley grains; this solvent removed 6.51% of the AR fraction of barley grains (Table 3), whereas wheat and rye lost only 1.20 and 1.57% of these antifungal compounds, respectively. Thus, out of two types of cereals, only kernels of rye retained a high level of these phenolic compounds and consequently prevented *F. culmorum* from enhancing its negative effects on cereal seedlings. In addition, our results follow the previous evidence of the localization of resorcinolic lipids mainly in the deeper-than-cuticle pericarp layers of caryopsis bran (fruit and grain coats) [7,8].

## 4. Conclusions

Our results markedly showed that the self-defense of cereal seedlings against diseases caused by *Rhizoctonia* fungi has been associated with the presence of a small amount of 5-*n*-alk(en)ylresorcinols on the caryopsis surface. However, their protective role is markedly dependent on both cereal species and cultivars. Moreover, these results suggest that an appropriate breeding strategy for cereals, taking into account the content and composition of grain surface ARs, could improve the resistance of cereals to soil-borne fungal pathogens. We hope that our current study contributes to extending the knowledge of the protective function of cereal ARs during the early stages of their seedling growth.

**Author Contributions:** Conceptualization, E.G.M. and S.J.P.; methodology, E.G.M.; formal analysis, E.G.M.; investigation, E.G.M.; writing—original draft preparation, E.G.M.; writing—review and editing, E.G.M., M.P.O. and S.J.P.; visualization, E.G.M. and M.P.O. All authors have read and agreed to the published version of the manuscript.

**Funding:** The APC is financed by the Wroclaw University of Environmental and Life Sciences.

**Institutional Review Board Statement:** Not applicable.

**Informed Consent Statement:** Not applicable.

**Data Availability Statement:** All data are contained within the article.

**Conflicts of Interest:** The authors declare no conflict of interest.

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
