# Peer review of "The Role of Resorcinolic Lipids of Caryopsis Surface in the Process of Cereal Infection by Rhizoctonia solani and Fusarium culmorum"

_applsci, doi:10.3390/app12157735_

Round 1

Reviewer 1 Report

The evaluated manuscript deals with the relationship between the presence of resorcinolic lipids on the surface of cereal grains and the susceptibility of their seedlings to infection by Rhizoctonia solani or Fusarium culmorum. The results of manuscript showed that self-defense of cereal seedlings against disease caused by Rhizoctonia and Fusarium fungi has been associated with the presence of a small amount of 5-n-alk(en)ylresorcinols on the surface of their kernels. Authors also state, that their protective role depends on both cereal species and cultivar, their various resistance to fungal pathogens. Manuscript with its structure and content corresponds to scientific work. In this work were cited 31 literary sources. Abstract and key words match the issue presented in the text. I do not have serious comments to the article and I recommend manuscript “The Role of Resorcinolic Lipids of Kernel Surface in the Process of Cereal Infection by Rhizoctonia Solani and Fusarium Culmorum” for publishing in Applied sciences Journal.

Author Response

Please see the attachment:)

Author Response

Please see the attachment:)

Reviewer 3 Report

Dear colleagues,

This review is concerning a research work entitled “The Role of Resorcinolic Lipids of Kernel Surface in the Process of Cereal Infection by Rhizoctonia Solani and Fusarium Culmorum, by Elżbieta G. Magnucka, Małgorzata P. Oksińska and Stanisław J. Pietr. As detailed data and interpretation, I recommend it for an international audience in this journal, however several points have to be precised and a minor revision is requested.

Please notice that in order to bring a broad audience to this article and to this journal, for specialists and non-specialists, the five major points of my comments (at the beginning) are very important (mandatory…) for a suitable value of the article. Minor points are also enhanced at the end of this review.

I deeply hope to see this good article published soon,

The five major points are:

11-    Although I am not English native speaker, this text is fully understandable, however some parts are not very fluent and a precise re-writing would make it much more english-standardized and more attractive for this international journal.

22-     As I am involved in plant taxonomy I am very sensible to correct taxa names, which should be inserted at least the first time they appear in the text. So in the introduction, lines 40-42, insert here the latin names (and author(s)) of the plants cited instead of in 2.2 and 2.4; use also this rule for all other plants like quinoa and others cited in different parts of the text…; do the same for fungi and cultivars and fungi strains (for these two latter organisms, if there is no author put at least the reference of the article or other document where they first appeared. Use international Plant Names Index (IPNI) https://www.ipni.org/) for plants, or equivalent; for fungi use the International Code of Nomenclature for algae, fungi, and plants (https://www.iapt-taxon.org/nomen/main.php), or equivalent.

33-     In order to be more attractive especially for non-specialists, a figure including detailed photos of the plants (infected or not) is necessary.

44-     All through the text, although it is frequently used in other papers, "kernel" is not very clear as we do not understand the difference(s) with caryopsis (which is one-seeded fruit in which the ovary wall is united with the seed coat); moreover remove "seed" from the text (especially in 2.3…) as Poaceae have caryopsis and not seeds; consider also that "grain" is commonly used in many papers however it has no precise botanical meaning. Please check for all other families of plants cited in the text if you can use "seed" or not (e.g., quinoa?).

55-     References already taken in account by the authors are of real interest, however checking briefly in the word of science WOS with the key-words of the abstract, other articles appear and references should be once more selected and used (if relevant…) in order to provide a larger view of this interesting research. Among these are the followings:

[1-16]

1.         Hoffmann, F.; Wenzel, G. Selfcompatibility in microspore-derived doubled-haploid rye lines and single grain selection for alkylresorcinol content. Theor Appl Genet 1981, 60, 129-133.

2.         Gohil, S.; Pettersson, D.; Salomonsson, A.-C.; Åman, P. Analysis of alkyl- and alkenylresorcinols in triticale, wheat and rye. Journal of the Science of Food and Agriculture 1988, 45, 43-52.

3.         Deszcz, L.; Kozubek, A. Higher cardol homologs (5-alkylresorcinols) in rye seedlings. Biochimica et Biophysica Acta (BBA) - Molecular and Cell Biology of Lipids 2000, 1483, 241-250.

4.         Abdel-Sattar, M.; El-Sherif, E.; El-Marzouky, H.; Mahmoud, M. BIOCHEMICAL RESPONSE IN POTATO CULTIVARS TO INFECT WITH Rhizoctonia solani THE CAUSAL OF STEM CANKER AND BLACK SCURF DISEASE IN EGYPT. Journal of Plant Protection and Pathology 2009, 34, 6941-6951.

5.         Pilar, M.C.; Ortega, N.; Perez-Mateos, M.; Busto, M.D. Alkaline Phosphatase−Polyresorcinol Complex: Characterization and Application to Seed Coating. J Agr Food Chem 2009, 57, 1967-1974.

6.         Kubus, G.; Tłuscik, F. Alkyl resorcinols in grains from plants from the family Gramineae. Acta Societatis Botanicorum Poloniae 2014, 52, 223-230.

7.         Chen, C.-Y.; Yang, T.-H.; Pan, C.-D.; Wang, X. Improved synthesis, X-ray structure, and antifungal activity of a sugar-psoralen conjugate: 4,4′-Dimethylxanthotoxol 2,3,4,6-tetra-O-Acetyl-β-D-glucoside. Journal of Carbohydrate Chemistry 2019, 38, 179-191.

8.         Landberg, R.; Hanhineva, K.; Tuohy, K.; Garcia-Aloy, M.; Biskup, I.; Llorach, R.; Yin, X.; Brennan, L.; Kolehmainen, M. Biomarkers of cereal food intake. Genes & Nutrition 2019, 14, 1-16.

9.         Sun, Y.; Yao, R.; Ji, X.; Wu, H.; Luna, A.; Wang, Z.; Jetter, R. Characterization of an alkylresorcinol synthase that forms phenolics accumulating in the cuticular wax on various organs of rye ( <i>Secale cereale</i> ). The Plant Journal 2020, 102, 1294-1312.

10.       Chrpová, J.; Orsák, M.; Martinek, P.; Lachman, J.; Trávníčková, M. Potential Role and Involvement of Antioxidants and Other Secondary Metabolites of Wheat in the Infection Process and Resistance to <i>Fusarium</i> spp. Agronomy 2021, 11.

11.       Ciccoritti, R.; Taddei, F.; Gazza, L.; Nocente, F. Influence of kernel thermal pre-treatments on 5-n-alkylresorcinols, polyphenols and antioxidant activity of durum and einkorn wheat. European Food Research and Technology 2021, 247, 353-362.

12.       Frølich, W.; Åman, P. Definition of Whole Grain and Determination of Content in Cereal Products; 2021.

13.       Magnucka, E.G.; Oksińska, M.P.; Pietr, S.J. Monitoring of changes in 5-n-alkylresorcinols during wheat seedling development. Zeitschrift für Naturforschung C 2021, 76, 67-70.

14.       Marklund, M.; Biskup, I.; Kamal‐Eldin, A.; Landberg, R. Alkylresorcinols and Their Metabolites as Biomarkers for Whole grain Wheat and Rye; 2021.

15.       Shahidi, F.; Danielski, R.; Ikeda, C. Phenolic compounds in cereal grains and effects of processing on their composition and bioactivities: A review. Journal of Food Bioactives 2021, 15.

16.       Zabolotneva, A.A.; Shatova, O.P.; Sadova, A.A.; Shestopalov, A.V.; Roumiantsev, S.A. An Overview of Alkylresorcinols Biological Properties and Effects. Journal of Nutrition and Metabolism 2022, 2022.

 As minor points:

1 in the introduction, for lines 35-39, provide a scheme of the caryopsis with the different parts involved, it will be more clear especially for non-specialists; moreover I am not sure that "layers" is the proper word for these different parts, please refer to classical botanical descriptions and terms of caryopsis;

2 in the introduction line 49, for quinoa which is a Chenopodiaceae, be sure that the right term used is “seed” as in this case the seed is enclosed in the  fruit which is an achene (indehiscent fruit with one seed…); moreover the term “pseudocereal” is a little bit ambiguous for me as “cereals” is almost entirely devoted to Poaceae;

3 in the introduction line 57 put "of the genera" instead of "of the genus";

4 at the end of 2.4, line 101put "was measured" instead of "was estimated"?

5 for the title of 2.5, I am not sure that "determination" is the proper word; see also the last sentence of 2.5;

6 all through the text, please check with the journal if for the references cited in your text, their number in brackets and authors (e.g. in 3.1 line 174 “Garcia and coworkers (31)”) are necessary; actually this is not at all homogeneous in the present text;

7 in 3.1, the last  sentence "Considerable discrepancies..." would need more discussion and potential explanation, if available;

8 for figure 2, explain shortly in the caption the meaning of small letters a b c d e f g... put also these small letters in bold (as the values), they are too difficult to read in the present state;

9 just before the conclusion point 4, line 332 I am not sure that the term “outer” cuticle is necessary as the cuticle is already known to be the outermost part of the caryopsis (just below the wax deposit);

10 in the conclusion point 4, the sentence "moreover, these results..." is too long and difficult to understand;

 11 for the list of references, the same number is used twice (e.g. 1.  1.), check in the recommendations to authors of this journal if it is correct;

12 check again one by one if all latin names are in italics (e.g. Rhizoctonia line 220).

Author Response

Please see the attachment:)
